# Fast Fallback Watermark Detection Using Perceptual Hashes

Hannes Mareen *, Niels Van Kets, Peter Lambert and Glenn Van Wallendael

imec—IDLab, Department of Electronics and Information Systems, Ghent University, Technologiepark-Zwijnaarde 122, 9052 Gent, Belgium; Niels.VanKets@UGent.be (N.V.K.); Peter.Lambert@UGent.be (P.L.); Glenn.Vanwallendael@UGent.be (G.V.W.)
* Correspondence: hannes.mareen@ugent.be

**Abstract:** Forensic watermarking is often used to enable the tracing of digital pirates that leak copyright-protected videos. However, existing watermarking methods have a limited robustness and may be vulnerable to targeted attacks. Our previous work proposed a fallback detection method that uses secondary watermarks rather than the primary watermarks embedded by existing methods. However, the previously proposed fallback method is slow and requires access to all watermarked videos. This paper proposes to make the fallback watermark detection method faster using perceptual hashes instead of uncompressed secondary watermark signals. These perceptual hashes can be calculated prior to detection, such that the actual detection process is sped up with a factor of approximately 26,000 to 92,000. In this way, the proposed method tackles the main criticism about practical usability of the slow fallback method. The fast detection comes at the cost of a modest decrease in robustness, although the fast fallback detection method can still outperform the existing primary watermark method. In conclusion, the proposed method enables fast and more robust detection of watermarks that were embedded by existing watermarking methods.

**Keywords:** watermarking; traitor tracing; perceptual hashing; content-based fingerprinting; watermark detection; fallback





## 1. Introduction

Forensic watermarking is used to track down digital pirates after they leak video content, i.e., to perform traitor tracing [1–3], i.e., each user receives a uniquely watermarked version of the video. When malicious users leak their version, they can be identified by means of the watermark. It is often a requirement for the watermark to be imperceptible, such that it does not decrease the quality of experience of honest users.

Most importantly, the watermark should be robust against attacks. This means that the watermark should still be detectable after pirates performed video processing manipulations in an attempt to delete the watermark. Typically, there is a trade-off between imperceptibility and robustness: a less perceptible watermark is typically easier to delete than a more perceptible one. Due to this trade-off, the watermark always has a limited robustness. If the quality is decreased sufficiently, then the watermark becomes undetectable. Additionally, existing methods can be vulnerable to targeted attacks that delete the watermark while retaining a high video quality.

To increase the level of robustness of existing watermarking techniques, our previous work proposed a fallback method that aimed to still detect the watermark when the traditional detection methods fail [4]. Rather than using the primary watermark that was embedded by the traditional method, the fallback uses a secondary watermark that consists of compression artifacts that are automatically created during compression of a primary-watermarked video. This fallback method demonstrated a great increase in robustness. For example, even when all primary watermark information was deleted, the secondary watermark could still be detected. Moreover, stronger robustness is achieved for standard signal processing attacks, such as recompression and noise addition.

Although the previously proposed fallback watermarking method effectively improves the robustness of existing techniques using secondary watermarks, the detection process is non-blind, i.e., a leaked video (segment) is compared to all distributed watermarked videos. This means that access to all watermarked videos is required for fallback detection. Additionally, comparing a leaked video to all watermarked videos is slow when there are many watermarked videos. Thus, this fallback method is not scalable.

This paper proposes to represent the secondary watermarks using perceptual hashes to make the fallback method more scalable. Perceptual hashes, also called content-based fingerprints, represent a video in a much smaller hash that is robust to common signal processing transformation. Using perceptual hashes with sizes between 570 bytes and 20,026 bytes for a 10-s video segment, the fallback detection method is sped up with a factor between 26,000 and 92,000.

In summary, the main contributions of this paper are the following:

- This is the first work that uses perceptual hashes to improve the performance of forensic watermarking methods.
- This work demonstrates that using perceptual hashes of secondary watermarks still allows for robust fallback detection.
- By calculating the perceptual hashes prior to detection, the fallback method is sped up with a factor between approximately 26,000 and 92,000, and does not require access to the watermarked videos during detection.

The remainder of this paper is organized as follows. First, Section 2 briefly discusses the relevant state of the art of watermarking and perceptual hashing. Then, Section 3 describes the proposed fast fallback watermarking method using perceptual hashes. Subsequently, the experimental results are given and analyzed in Section 4. Next, the limitations are discussed in Section 5, and a practical example is given in Section 6. Finally, the main findings of this paper are discussed and concluded in Section 7.

## 2. State of the Art

### 2.1. Forensic Watermarking

Existing watermarking techniques can be grouped in three main categories, depending on when in the compression pipeline the watermark is embedded, namely (1) in the uncompressed domain (before compression), (2) in the loop (during compression), and (3) out of the loop (after compression). Additionally, watermarks can also be embedded in the uncompressed domain, after decompression, on the client's device.

Most watermarking techniques operate in the uncompressed domain, as this is codec independent. For example, many techniques are based on spread-spectrum watermark embedding, originally proposed by Cox et al. [5]. In such methods, the amplitude of certain transform coefficients is modulated according to the pseudorandomly generated watermark. Often, this is done in the discrete cosine transform (DCT) domain [5–7], although other transformations are commonly used as well [8,9]. For example, the dual-tree complex wavelet transform (DTCWT) method can be used to provide more robustness against geometric attacks [10–12]. In any case, a trade-off between imperceptibility and robustness is made by choosing either higher or lower-frequency coefficients: low-frequency coefficients are more perceptible, yet more robust. A main disadvantage of uncompressed-domain watermarking methods that are applied before compression is that every watermarked video needs to be separately compressed. This compression is an unintentional attack on the watermark and, more importantly, is a very complex and computationally expensive process. However, methods exist to lower the complexity of compression for watermarked videos [13], or to reduce the number of required watermarked videos [14]. In this way, these methods can be deployed at scale in practice. To alternatively use uncompressed-domain watermarking methods in a scalable way, they can be applied on each individual client's device, after decompressing the received video [15,16]. However, such techniques decrease the overall security of the system, since the unwatermarked video will be temporarily stored on the client's device, and can thus potentially be accessed by malicious users.

Some watermarking methods are applied in the loop, i.e., during compression [17–22]. These methods are codec dependent and still have the disadvantage that they require every watermarked video to be compressed separately. Alternatively, a watermark can be embedded out of the loop, i.e., after compression [23–28]. Embedding watermarks directly in the compressed domain has the advantage that only the unwatermarked video needs to be compressed, and thus it is much more scalable. However, it is not straightforward to modify the compressed video without affecting the video quality or bit rate.

In general, all existing forensic watermarking techniques fail when exposed to certain attacks, because they need to make a trade-off between imperceptibility and robustness. For example, when the video quality is significantly decreased, the watermark cannot be detected anymore. Additionally, targeted attacks may be invented that delete a watermark with less impact on video quality.

### 2.2. Fallback Detection Using Secondary Watermark

To improve the robustness of existing uncompressed-domain watermarking methods, our previous work proposed a fallback detection method that aims to detect the watermark when the traditional watermarking method cannot [4]. First, it was demonstrated that a secondary watermark signal is automatically and indirectly created during compression of a video with a primary watermark that was embedded by an existing method. The secondary watermark consists of compression artifacts, and may be present in other regions and components than the primary watermark. For example, the DTCWT method of Asikuzzaman et al. embeds a primary watermark in certain DTCWT coefficients of the U channel of every frame of the video [10]. A grayscale attack that deletes the U and V channel is hence an effective attack that deletes all primary watermarking information. However, our previous work demonstrated that although this primary watermark is only present in the U channel, the secondary watermark is present in all channels. As such, the secondary watermark can survive such targeted attacks.

The secondary watermark is exploited in the fallback detection method to improve the robustness of existing watermarking techniques, i.e., the secondary watermark of a leaked video is compared to the secondary watermarks of all watermarked versions of the video. This is done by calculating the correlation coefficient (cc) between the leaked and watermarked videos, frame by frame [29]. The correlation coefficient (cc) is an extension of the normalized correlation (nc) and is defined in (1).

$$
\mathrm{nc}(a_f, w_{i,f}) = \sum_{x}^{W} \sum_{y}^{H} \frac{a_{f,x,y}}{|a_f|} \cdot \frac{w_{i,f,x,y}}{|w_{i,f}|},
$$
$$
\mathrm{cc}(a_f, w_{i,f}) = \mathrm{nc}(a_f - \bar{a}_f, w_{i,f} - \bar{w}_{i,f})
\tag{1}
$$

In the equation, $a_f$ and $w_{i,f}$ are the frames of width $W$ and height $H$, representing the $f^{\mathrm{th}}$ frame of the attacked and compressed watermarked video (using primary watermark $i$), respectively. Additionally, $a_{f,x,y}$ and $w_{i,f,x,y}$ represent the pixel at position $(x, y)$ within those frames. Furthermore, $|a_f|$ and $|w_{i,f}|$ represent the Euclidean length of the vectors $a_f$ and $w_{i,f}$, respectively, and $\bar{a}_f$ and $\bar{w}_{i,f}$ represent their mean pixel values. In other words, the uncompressed video signals are compared pixel by pixel. For simplicity, only the luminance pixel values were used.

The correlation with one watermarked video should be significantly higher than the correlation with all other watermarked videos. This indicates that the receiver of this watermarked video leaked the video. Therefore, outlier detection is performed by calculating the z-score for each correlation value. The z-score normalizes the correlation values, such that a high z-score indicates the corresponding watermark's presence, whereas a z-score close to zero indicates it is absent. In practice, the z-score is compared to a threshold that is calculated for a certain false-positive (FP) probability.

Most importantly, comparing the uncompressed video signal (or secondary watermark signal) of the leaked video to the uncompressed signals from all watermarked videos is the

most complex step in the fallback detection process. When there are many watermarked videos, these comparisons take the most time. Moreover, this method requires access to all watermarked videos. To solve these issues, Section 3 proposes a fast adaptation of this fallback method using perceptual hashes.

*2.3. Perceptual Hashes*

Perceptual hashing, also called content-based fingerprinting methods, are typically used for applications such as video copy detection, i.e., it is used to automatically detect whether two videos are perceptually equal. For example, it can be used to scrape the Internet and automatically detect whether the found videos are present in a certain database (e.g., to find copyright-protected videos). They do this by extracting a perceptual hash or fingerprint of the video that is robust and discriminant. Robust means that two videos that have the same content should have an approximately equal hash, even when they have undergone different content-preserving operations such as recompression. Discriminant means that two perceptually different videos should have significantly different hashes. Although most perceptual hashing research was performed on images [30–33], video copy detection has been investigated as well [34–40].

There are a variety of strategies to perform perceptual hashing. For example, a temporal fingerprint can be created using the shot boundaries [35]. This can be efficient for detecting a copy of a full-length movie, but does not work well in short video segments. Another way to calculate perceptual hashes is using global descriptors of the video based on motion, color and spatio-temporal distribution of pixel intensities [41]. Alternatively, Coskun et al. proposed hashing methods based on the three-dimensional (3D) DCT [34]. Similarly, Esmaeili et al. used the DCT for hashing, yet using two-dimensional (2D) DCT on temporally informative representative images (TIRIs) instead of a 3D DCT on the 3D video signal [36]. A TIRI is a 2D signal that can be seen as a superframe that contains both spatial and temporal information of a very short video segment. The concept of TIRI-DCT hashes is more thoroughly explained in Section 3.1.

Most importantly, perceptual hashes are compact, easy to compute and fast to search for in a large database. These are properties that are missing but desired in the slow fallback watermark detection method of our previous work [4]. Thus, it is interesting to combine forensic watermarking and perceptual hashing, which is proposed in Section 3.

## 3. Materials and Methods

The core idea of the proposed method is to perform fallback watermark detection using perceptual hashes of secondary watermarks. This is much faster in comparison with using the uncompressed secondary watermark signals, which is done in our previous work [4]. Additionally, perceptual hashes require much less storage.

Figure 1 shows a high-level diagram of the proposed fast watermarking technique. The leaked video $l$ and each watermarked video $w_i$, $i \in \{1, 2, \ldots, N\}$ (with $N$ the number of watermarked videos) are perceptually hashed, with the unwatermarked video $u$ as secondary input. Then, all these perceptual hashes are given as input of the fast fallback detection method. Finally, the method outputs a decision about each watermark's presence. The perceptual hashing and fast fallback detection method are described in Section 3.1 and Section 3.2, respectively. Subsequently, the computational complexity is discussed in Section 3.3.

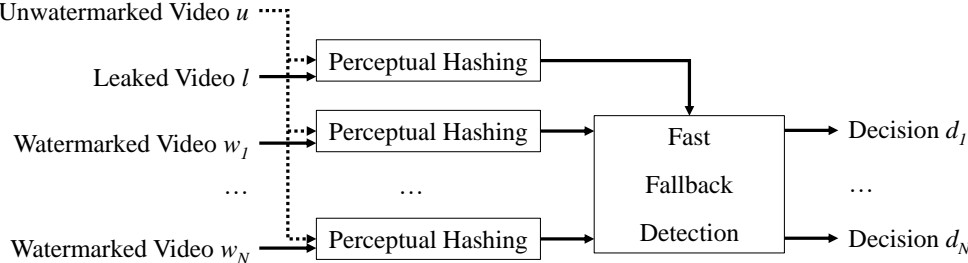

**Figure 1.** High-level diagram of the proposed fast fallback method using perceptual hashing.

### 3.1. Perceptual Hash of Secondary Watermark

This section describes the perceptual hashing method that is used to fingerprint the secondary watermark signal. This paper incorporates the TIRI-DCT algorithm by Esmaeili et al. to transform a secondary watermark signal to a perceptual hash [36], although another perceptual hashing method could be used as well. Please note that the perceptual hashing methods were typically made for copy detection of video content, as explained in Section 2.3, whereas the proposed method targets watermark detection. In other words, the original method is applied on an entirely different use case.

Figure 2 shows a high-level diagram of the proposed perceptual hashing method. The method takes a watermarked (potentially attacked-and-leaked) video $w$ and the corresponding unwatermarked video $u$ as input. First, both videos are preprocessed. Then, the preprocessed videos are split into smaller segments, and each segment is transformed to a TIRI (i.e., a superframe that contains both spatial and temporal information of the segment). Subsequently, the TIRIs from the unwatermarked video are subtracted from the TIRIs of the watermarked video to obtain TIRIs from the secondary watermark signals. Finally, a DCT-based hash is calculated from the TIRIs. The remainder of this section describes these individual steps of the perceptual hashing method in more detail. It should be noted that this hashing procedure can be performed on a short video segment, and not necessarily on the full-length video. For example, the experiments in Section 4 consider 10-s and 50-s video segments.

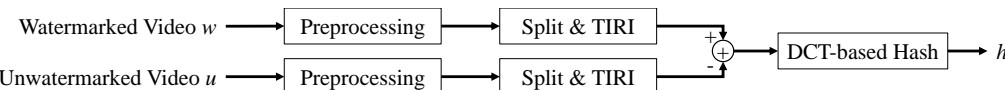

**Figure 2.** High-level diagram of the proposed perceptual hashing method.

The preprocessing of a video involves downsampling in both space and time. More specifically, the video is rescaled from a resolution of $W \times H$ to $W_d \times H_d$, and the framerate is reduced from $R$ to $R_d$ frames per second (fps). This reduces the total number of frames in the video segment from $F$ to $F_d$ frames. Additionally, for simplicity, only the luminance values of the pixels are used in the remainder of the proposed method.

Then, the downsampled watermarked and unwatermarked video are divided into (very) short overlapping segments of $J$ frames, with 50% overlap. For each very short segment, a TIRI $t$ is created as defined in (2). Let $t_{x,y}$ be the pixel value of the TIRI at position $(x, y)$, and let $s_{f,x,y}$ be the luminance pixel value at position $(x, y)$ of frame $f$ in the short video segment. The pixels of the TIRI are obtained as a weighted sum of the frames of the segment. Exponential weighting is used because it was observed that it best captures the temporal motion [36]. The weight factor $\gamma^f$ is used, where $f$ is the frame number in the short segment. By changing $\gamma$ from 0 to 1, the TIRI $t$ moves from selecting a single frame (i.e., high spatial information) to selecting all frames with equal weights (i.e., high temporal information).

$$t_{x,y} = \sum_{f}^{J} \gamma^f s_{f,x,y} \qquad (2)$$

Figure 3 visualizes the process of transforming a short (preprocessed) video segment of $J = 10$ frames to a single TIRI, with $\gamma = 0.5$. It can be seen that the TIRI shows both spatial information (the trees and background are clearly visible) and temporal information (the biker is blurry because it is in motion).

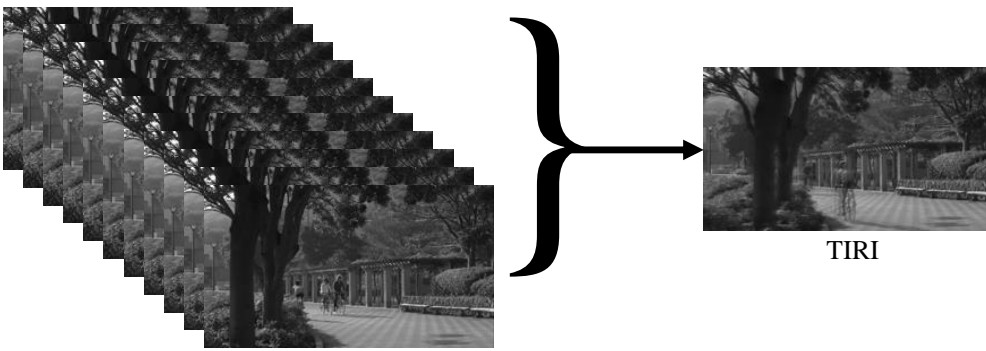

**Figure 3.** Visualization of transformation of short preprocessed video segment that contains multiple frames to a single TIRI.

After reducing the videos into TIRIs, the unwatermarked TIRIs are subtracted from the watermarked TIRIs. This subtraction was not done in the original perceptual hashing algorithm, since it was created to summarize the underlying video content. In contrast, the fallback method is not interested in the underlying content since it is the same for all watermarked videos. Instead, the fallback method wants to hash the secondary watermark signal. Therefore, we first subtract the unwatermarked TIRI from the watermarked TIRI.

Finally, the TIRIs are transformed into a DCT-based hash as shown in the diagram in Figure 4. First, the TIRI is split into overlapping blocks of size $2b \times 2b$ (with 50% overlap). Then, every block is transformed using the DCT (type II). Subsequently, some DCT coefficients are extracted. In the original TIRI-DCT method by Esmaeili et al., it was proposed to only extract the first horizontal and first vertical coefficients adjacent to the DC coefficient. We propose to expand this to extracting the $d \times d$ lowest-frequency DCT coefficients. Additionally, $d = 0$ signifies the special case in which only the first horizontal and first vertical coefficient are extracted. In this way, we allow the hash size to be increased more easily, which may result in better robustness performance. Afterward, all DCT coefficients of all blocks are concatenated into a single vector $c$ per TIRI. After extracting and concatenating the coefficients $c$ of a TIRI $t$, they are binarized. This is done by comparing each coefficient $c_i$ to the median $m$ of all coefficients. In other words, a binary perceptual hash $h$ is created as defined in (3).

$$h_i = \begin{cases} 1, & c_i \geq m \\ 0, & c_i < m \end{cases} \tag{3}$$

TIRI $t$ → Split in blocks → DCT → Extract coefficients → Concatenate → Binarize → Hash $h$

**Figure 4.** High-level diagram of the DCT-based hashing method that transforms a TIRI into a perceptual hash.

In short, perceptual hashing of a secondary watermark signal happens in the following way. First, the Y channel of the watermarked and unwatermarked video are downsampled and divided in (very) short video segments. Each of these segments are transformed to a TIRI. Then, the unwatermarked TIRIs are subtracted from the watermarked TIRIs, resulting in secondary watermark TIRIs. These TIRIs are split in overlapping blocks, and certain DCT coefficients blocks are extracted from each block. Subsequently, all extracted coefficients are

concatenated into a single vector. Finally, the vector of each TIRI is binarized by comparing each value to the median, resulting in a binary perceptual hash. This hash is much smaller than the secondary watermark signal.

### 3.2. Fast Fallback Detection Using Perceptual Hashes

After creating a perceptual hash of each secondary watermark and the leaked video using the procedure described in Section 3.1, the proposed fallback detection method can be performed, which is described in this section. The proposed method is based on the same concepts as the slow fallback detection method of previous work [4], yet it is much faster due to using perceptual hashes.

In the original fallback method of our previous work [4], the secondary watermark signal of the leaked video is compared to the secondary watermarks of all watermarked videos, using the correlation coefficient as defined in (1). In contrast, in the proposed adapted fallback method, we compare the corresponding perceptual hashes. To compare the perceptual hash of a leaked video to the perceptual hash of a watermark, we use the accuracy measure, i.e., the number of equal bits divided by the total number of bits in the hash. This can be implemented using logical XOR operations, and hence is very fast. As such, the accuracy is calculated between the perceptual hash of the extracted secondary watermark of the leaked video, and the perceptual hashes of the secondary watermarks of all watermarked videos. This results in the accuracies $a_i$, $i \in \{1, 2, \ldots, N\}$, with $N$ the number of watermarked videos.

The accuracy corresponding to the watermark that is present in the leaked video is higher than the accuracies corresponding to all other watermarks. An example of this is shown in Figure 5. To create the figure, a recompressed version of watermarked video with ID-10 is considered the leaked video. The figure shows the accuracies between the perceptual hash of the extracted secondary watermark of this leaked video and the perceptual hashes of the secondary watermarks of all 20 watermarked videos. The video is the *ParkScene* sequence, which has a resolution of $W \times H = 1920 \times 1080$ and contains $F = 240$ frames at $R = 24$ fps [42]. The perceptual hashing algorithm is the one described in Section 3.1 and uses the following parameters: $W_d = 256, H_d = 144, R_d = 10, J = 10, \gamma = 0.5, b = 16, d = 4$. It can be observed that the accuracy corresponding to ID-10 is approximately 0.73, whereas the accuracies corresponding to all other watermarks are lower, i.e., approximately 0.675.

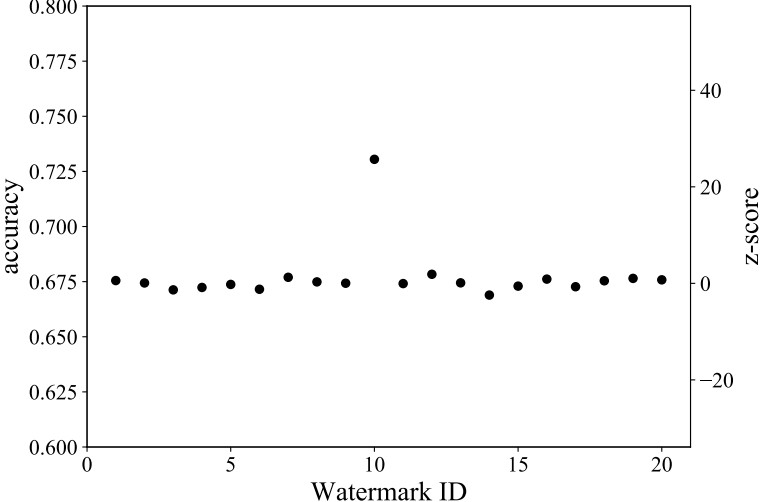

**Figure 5.** The accuracies and corresponding z-scores when comparing the perceptual hash of the extracted secondary watermark of an attacked version of watermarked video with ID-10 to the perceptual hashes of the secondary watermarks of all 20 watermarked videos.

To better analyze the accuracies, outlier detection is performed by calculating the z-score $z_i$ for each accuracy value $a_i$, as defined in (4). The z-score $z_i$ indicates the number of standard deviations $\sigma_A$ that an accuracy value $a_i$ differs from the mean $\mu_A$ of the distribution of accuracy values corresponding to absent watermarks. For simplicity, it is estimated that all watermarks are absent, except for the one corresponding to the highest accuracy.

$$z_i = \frac{a_i - \mu_A}{\sigma_A} \tag{4}$$

The z-score normalizes the accuracy values corresponding to absent watermarks. As such, the distribution of z-scores corresponding to absent watermarks has a zero mean and unit variance. In contrast, the z-score corresponding to a present watermark is much larger than zero. Because of this contrasting difference, it is possible to compare the z-score to a threshold to decide about a watermark's presence. The Gaussian method is used to estimate this threshold $T_F$ for a certain FP probability $P_{fp}$, and is defined in (5). In the equation, $\mathrm{erfc}^{-1}$ is the inverse of the complimentary error function. For example, for a FP probability of $P_{fp} = 10^{-6}$, the corresponding threshold is $T_F \approx 4.8$.

$$T_F = \sqrt{2}\,\mathrm{erfc}^{-1}(2P_{fp}) \tag{5}$$

Finally, using the obtained z-score and threshold, a decision $d_i$ of watermark presence or absence can be made using (6). In other words, a watermark's presence is detected when the corresponding z-score is larger than this threshold.

$$d_i = \begin{cases} \text{Present}, & \text{if } z_i \geq T_F \\ \text{Absent}, & \text{otherwise} \end{cases} \tag{6}$$

In Figure 5, the z-scores corresponding to the accuracies are shown on the secondary vertical axis. One can observe that the z-score of watermark ID-10 is approximately 25, which is significantly higher than the threshold $T_F \approx 4.8$, and hence watermark ID-10 is detected in the leaked video. In contrast, the other z-scores vary between $-2$ and $+2$, which are under the threshold, meaning that the other watermarks are not detected.

It should be noted that the secondary watermark is a zero-bit watermark. Hence, it cannot be used to embed a specific multi-bit payload. Instead, only its presence or absence can be detected, and subsequently linked to a certain primary watermark (which may or may not be a multi-bit watermark). Subsequently, the primary watermark can be a multi-bit message, or another zero-bit watermark that is linked to, for example, a certain user.

In summary, the fallback detection method works in the following way. First, the perceptual hashes are calculated from the extracted secondary watermark of the leaked video, as well as from the secondary watermarks of all watermarked videos. Then, the accuracy is calculated between the perceptual hash of the leaked video with the perceptual hashes of all watermarked videos. These accuracies are transformed to z-scores, which are subsequently compared to a threshold $T_F$ to detect a watermark's presence.

### 3.3. Theoretical Complexity Analysis

The main novelty of the proposed method is that fallback detection is sped up using perceptual hashes instead of the uncompressed secondary watermark signals. This section briefly compares the theoretical computational complexity of the slow fallback method of previous work with the proposed fast fallback method.

The slow fallback method calculates the correlation coefficient between the uncompressed leaked and watermarked videos, using the formula given in (1). In other words, a pixel-by-pixel comparison is performed between the leaked video and each watermarked video. Hence, watermark detection has a computational complexity of $\Theta(N \cdot F \cdot W \cdot H \cdot B)$, in which $N$ is the number of watermarked videos, $F$ is the number of frames of the video segment, $W$ and $H$ are the width and height, and $B$ is the number of bits per pixel.

The fast fallback transforms each video to a perceptual hash with a much smaller size. In short, this is done by downsampling the video, dividing it in overlapping segments and overlapping blocks, selecting certain DCT coefficients of each block, and binarizing the resulting vector. This results in a perceptual hash with a length $L$, defined in (7).

$$
L = \begin{cases} \dfrac{F_d - \frac{J}{2}}{\frac{J}{2}} \cdot \dfrac{W_d - b}{b} \cdot \dfrac{H_d - b}{b} \cdot 2, & \text{if } d = 0 \\[2ex] \dfrac{F_d - \frac{J}{2}}{\frac{J}{2}} \cdot \dfrac{W_d - b}{b} \cdot \dfrac{H_d - b}{b} \cdot d^2, & \text{if } d \geq 1 \end{cases}
\tag{7}
$$

The fast fallback method compares the short hashes of length $L$ using the accuracy measure. This can be efficiently implemented using the logical XOR operator. Hence, the computational complexity is $\Theta(N \cdot L)$. In other words, the complexity is still linear in the number of watermarked videos, but has a significant speed-up when $L$ is much smaller than $F \cdot W \cdot H \cdot B$.

As an example, consider a video segment with resolution $W \times H = 1920 \times 1080$ pixels, $F = 500$ frames at $R = 50$ fps, and $B = 8$ bits per luminance pixel value. Assume the following parameters are used for perceptual hashing: $W_d = 256, H_d = 144, R_d = 10,$ $J = 10, b = 8, d = 4$. This results in hashes of size $L = \dfrac{100 - 5}{5} \cdot \dfrac{256 - 8}{8} \cdot \dfrac{144 - 8}{8} \cdot 4^2 =$ $160,208$ bits $= 20,026$ bytes. As a result, for these parameters, we obtain a speed-up of $\dfrac{N \cdot F \cdot W \cdot H \cdot B}{N \cdot L} \approx 51,772$, in theory.

Lastly, note that constant factors are ignored in the asymptotic complexity notations, and hence the actual speed-up may differ. The practically observed speed-up is measured in Section 4.4.

## 4. Results

This section experimentally evaluates the proposed fallback method based on perceptual hashes of secondary watermarks. First, Section 4.1 describes the experimental setup. Then, the perceptibility of the primary and secondary watermark are briefly discussed in Section 4.2. Next, Section 4.3 evaluates the robustness, and Section 4.4 analyzes the time measurements.

### 4.1. Experimental Setup

For comparison purposes, the experimental setup is equal to the setup used in the slow fallback method of our previous work [4]. That is, five 10-s sequences of resolution $1920 \times 1080$ pixels are used: *BQTerrace, Cactus, Kimono1, ParkJoy*, and *ParkScene* [42]. These contain 600, 500, 240, 500, and 240 frames, respectively.

As a primary watermarking method, the DTCWT method of Asikuzzaman et al. is used [10]. The parameters of the DTCWT method are the same as those used by Asikuzzaman et al. and in the fallback method of previous work, i.e., the step value $\Omega$ was set to 6 and 1 at the watermark embedder and detector, respectively, and the strength $\alpha_{DTCWT}$ was set to 36. Additionally, the threshold was set to $T_{DTCWT} = 0.0526$, which corresponds to $P_{fp} = 10^{-6}$ (using a $k$-dimensional watermark with $k = 8160$). Please note that the DTCWT method embeds a primary watermark in the U channel of the video, whereas the proposed fallback method only uses the luminance information of the Y channel as a secondary watermark.

Twenty watermarked versions of each test sequence were created (ID-1 to ID-20). Then, each of these 20 versions were compressed using the *x265*-encoder, which uses the High Efficiency Video Coding (HEVC) standard. More specifically, four constant rate factors (CRFs) were used that automatically vary the quantization parameters (QPs): 22, 27, 32, and 37, further denoted as $QP_w$ (where $w$ stands for *watermark*). During compression, the intra period was set to the video segment length and the other encoding parameters were set to their defaults.

Each watermarked video is attacked using a grayscale-and-recompression attack, i.e., the U and V channel are deleted from the video to delete the primary watermark, followed by a recompression in an effort to obstruct the secondary watermark. The recompression is performed with the same encoder as the initial encoding (i.e., the *x265*-encoder), using default parameters and 6 different QPs: 22, 27, 32, 37, 42, and 47, further denoted as $QP_a$ (where *a* stands for *attack*). For clarity, the difference between $QP_w$ and $QP_a$ is that the former is the QP used during initial compression of the watermarked video, which is typically performed by the video distributor. In contrast, $QP_a$ is the re-encoding performed by the attacker, before leaking the video. Please note that a recompression with $QP_a = 22$ retains a high quality, whereas $QP_a = 47$ severely lowers the quality to the point that it is unpleasant to watch.

To perform the proposed fallback detection, the following fixed parameters are used for the proposed perceptual hashing algorithm: $W_d = 256, H_d = 144, R_d = 10, J = 10, \gamma = 0.5$. This means that the test sequences are downsampled from a resolution of $1920 \times 1080$ with a framerate between 24 and 60 fps, to a resolution of $256 \times 144$ with a fixed framerate of 10 fps. Moreover, the downsampled secondary watermark signals are segmented in very short overlapping segments of 10 frames (i.e., 1 s), which are used to calculate the TIRIs. Additionally, the block size *b* is varied between 8 and 16, and the number of extracted DCT coefficients *d* is set to either 0, 2, or 4.

*4.2. Perceptibility*

For completeness, this section briefly reports on the perceptual transparency of the primary and secondary watermark. It should be stressed that the primary watermark embedding method is not adapted, and is applied as proposed in the state of the art. Moreover, the secondary watermark is created automatically during video compression. The proposed fallback detection method is hence applied without changing the original watermark embedding and compression process.

To measure the perceptibility of the watermarks, the Peak Signal-to-Noise Ratio (PSNR) measure is used. The average PSNR between the luminance channels (further denoted as $PSNR_Y$) of the unwatermarked original video and the primary-watermarked video is infinity. This is as expected, since the DTCWT method embeds a watermark in the U channel, i.e., it leaves the luminance channel untouched. The average PSNR between the corresponding U channels (denoted as $PSNR_U$) is approximately 37.92 decibel (dB).

During compression of the primary-watermarked video, the secondary watermark is automatically introduced by the video encoder. Thus, to measure the perceptibility of the secondary watermark by calculating the average $PSNR_Y$ between the uncompressed watermarked video and the compressed watermarked video. For a fair comparison, the average $PSNR_Y$ is also calculated between the uncompressed unwatermarked video and the compressed unwatermarked video. These results are given in Table 1. It can be observed that the $PSNR_Y$ of the compressed watermarked and compressed unwatermarked video are approximately equal, for equal $QP_w$ values. This demonstrates that the secondary watermark is not more perceptible than the compression artifacts that are present in a compressed unwatermarked video. Additionally, it can be observed that larger $QP_w$ values result in lower $PSNR_Y$ values, which suggests that videos compressed with larger $QP_w$ values have a more robust secondary watermark.

**Table 1.** Perceptibility results of secondary watermark, in comparison with compression artifacts in unwatermarked videos.

| $QP_w$ | Average $PSNR_Y$ (dB) | |
| --- | --- | --- |
| | Unwatermarked | Watermarked |
| 22 | 40.78 | 40.76 |
| 27 | 38.42 | 38.42 |
| 32 | 35.86 | 35.87 |
| 37 | 33.29 | 33.26 |

### 4.3. Robustness

To measure the robustness, the false-negative rate (FNR) is calculated. The FNR is defined in (8), and represents the fraction of false-negative (FN) detections, i.e., the fraction of watermarks that were not detected yet were present in the video under investigation. Additionally, the FNR is related to the true-positive rate (TPR) as defined in (9).

$$\text{FNR} = \frac{\text{\#FN Detections}}{\text{Total Number of Detections}} \tag{8}$$

$$\text{TPR} = 1 - \text{FNR} \tag{9}$$

First, Table 2 gives the robustness results of the primary watermarking method (DTCWT), as well as the results of the slow fallback method of our previous work which compares the uncompressed secondary watermark signals rather than perceptual hashes. One can observe that the primary watermarking method fails to provide robustness for all $QP_a$ values, since the U channel (that contains the primary watermark) is removed. In contrast, the slow fallback method provides better robustness and only reports a nonzero FNR for $QP_w = 22$ and $QP_a = 47$. It should be stressed that this robustness comes at the cost of requiring the watermarked videos, and needing to compare all these watermarked videos to the leaked video. In an uncompressed form, these 10-s videos are between 711 MB and 1.73 GB large. Hence, comparing these takes a long time, as measured in-depth in Section 4.4.

**Table 2.** Robustness results for the primary watermarking method and the slow fallback method of previous work.

| $QP_w$ | | False-Negative Rate * (%) | | | | | | | | | | | |
|---|---|---|---|---|---|---|---|---|---|---|---|---|---|
| | $QP_a =$ | 22 | 27 | 32 | 37 | 42 | 47 | 22 | 27 | 32 | 37 | 42 | 47 |
| | | DTCWT [10] | | | | | | Slow Fallback [4] | | | | | |
| 22 | | 100 | 100 | 100 | 100 | 100 | 100 | 0 | 0 | 0 | 0 | 0 | 13 |
| 27 | | 100 | 100 | 100 | 100 | 100 | 100 | 0 | 0 | 0 | 0 | 0 | 0 |
| 32 | | 100 | 100 | 100 | 100 | 100 | 100 | 0 | 0 | 0 | 0 | 0 | 0 |
| 37 | | 100 | 100 | 100 | 100 | 100 | 100 | 0 | 0 | 0 | 0 | 0 | 0 |

* A 0% FNR means correct detection of all watermarks in all tested sequences.

The proposed fast fallback method uses perceptual hashes, which require much less storage and are much faster to compare. The robustness results when using various parameters for the perceptual hashing algorithm are given in Table 3. Additionally, the table gives the size of a single perceptual hash in bytes (B). First, one can observe that using a smaller block size ($b$) results in a larger hash size, and consequently also in better robustness. Secondly, extracting more DCT coefficients (i.e., a larger parameter $d$) also results in a larger hash size, and better robustness. When using a relatively small hash of 570 bytes for a 10-s video segment, i.e., when $d = 0$ and $b = 16$, the fallback is not very robust, i.e., nonzero FNRs are only observed when the recompression attack does not significantly decrease the quality of the video (i.e., the $QP_a$ is relatively low). In contrast, when using larger hashes of 20,026 bytes, i.e., when $d = 4$ and $b = 8$, the robustness performance is much better, i.e., the fallback can resist recompression attacks that more-significantly decrease the quality of the video, especially when it was initially compressed with a higher $QP_w$. Thus, there is a clear trade-off between storage and robustness: using larger hashes results in a better robustness.

**Table 3.** Robustness results for the proposed fast fallback method using perceptual hashes, as well as the size of the perceptual hashes in bytes (B), when using short 10-s sequences.

| $QP_w$ | $QP_a =$ | 22 | 27 | 32 | 37 | 42 | 47 | 22 | 27 | 32 | 37 | 42 | 47 |
|---|---|---|---|---|---|---|---|---|---|---|---|---|---|
| | | \multicolumn{12}{c}{$d = 0$} | | | | | | | | | | |
| | | \multicolumn{6}{c}{$b = 16$ (570 B)} | | | | | \multicolumn{6}{c}{$b = 8$ (2508 B)} | | | | | |
| 22 | | 0 | 40 | 99 | 100 | 100 | 100 | 0 | 0 | 60 | 100 | 100 | 100 |
| 27 | | 0 | 0 | 40 | 99 | 99 | 100 | 0 | 0 | 0 | 60 | 100 | 100 |
| 32 | | 0 | 0 | 0 | 30 | 99 | 100 | 0 | 0 | 0 | 0 | 58 | 98 |
| 37 | | 0 | 0 | 0 | 0 | 37 | 95 | 0 | 0 | 0 | 0 | 0 | 42 |
| | | \multicolumn{12}{c}{$d = 2$} | | | | | | | | | | |
| | | \multicolumn{6}{c}{$b = 16$ (1140 B)} | | | | | \multicolumn{6}{c}{$b = 8$ (5016 B)} | | | | | |
| 22 | | 0 | 17 | 97 | 100 | 100 | 100 | 0 | 0 | 51 | 100 | 99 | 100 |
| 27 | | 0 | 0 | 13 | 92 | 100 | 100 | 0 | 0 | 0 | 39 | 100 | 100 |
| 32 | | 0 | 0 | 0 | 18 | 93 | 100 | 0 | 0 | 0 | 0 | 70 | 99 |
| 37 | | 0 | 0 | 0 | 0 | 23 | 92 | 0 | 0 | 0 | 0 | 0 | 26 |
| | | \multicolumn{12}{c}{$d = 4$} | | | | | | | | | | |
| | | \multicolumn{6}{c}{$b = 16$ (4560 B)} | | | | | \multicolumn{6}{c}{$b = 8$ (20,026 B)} | | | | | |
| 22 | | 0 | 0 | 34 | 94 | 100 | 100 | 0 | 0 | 0 | 68 | 100 | 100 |
| 27 | | 0 | 0 | 0 | 36 | 98 | 100 | 0 | 0 | 0 | 0 | 74 | 99 |
| 32 | | 0 | 0 | 0 | 0 | 42 | 97 | 0 | 0 | 0 | 0 | 0 | 63 |
| 37 | | 0 | 0 | 0 | 0 | 0 | 25 | 0 | 0 | 0 | 0 | 0 | 1 |

\* A 0% FNR means correct detection of all watermarks in all tested sequences.

In Table 3, one can additionally observe that watermarked videos compressed with a high $QP_w$ value are robust to recompression attacks with higher $QP_a$ values. This was also observed in previous work, and can be ascribed to higher $QP_w$ values introducing more compression artifacts. Since the secondary watermark is represented by compression artifacts, it is more robust when the video is initially compressed with a lower quality setting.

To better compare the fast and slow fallback method, Figure 6 shows the receiver operating characteristic (ROC) curves for $QP_w = 22$ and $QP_a = 47$. An ROC curve compares two operating characteristics, namely the true-positive rate and false-positive probability. Because the ROC curve of the fast fallback method is hard to thoroughly inspect using a linear scale in Figure 6a, the same curves are plotted on a logarithmic scale in Figure 6b. The considered attack corresponds to the upper-right FNR value in Table 2; for a FP probability of $P_{fp} = 10^{-6}$, the slow fallback method has a 13% FNR (i.e., an 87% TPR). The fast fallback method is used with parameters $d = 4$ and $b = 8$, i.e., the same as in the lower-right part of Table 3. For a FP probability of $P_{fp} = 10^{-6}$, the fast fallback method has a 100% FNR (i.e., a 0% TPR). From Figure 6, can observe that the slow fallback method has much better performance than the fast fallback method, for this considered attack. For example, when the FP probability is set to $P_{fp} = 10^{-3}$, the slow fallback method has a 100% TPR, whereas the corresponding TPR of the fast fallback method is only 7%. Additionally, the Equal Error Rate (EER) line is plotted with the ROC curves, which shows where the FP probability is equal to the FNR (and FNR $= 1 - $ TPR). For the fast fallback method, the EER line intersects approximately at $P_{fp} \approx 0.65$, TPR $\approx 0.35$. In contrast, for the slow fallback method, the EER is approximately $P_{fp} \approx 10^{-3}$ (corresponding to TPR $\approx 1 - 10^{-3}$). It should be noted though that the performance of both fallback methods is much better for less-severe attacks, as observed in Tables 2 and 3. Thus, in short, the slow fallback method demonstrates better robustness than the fast fallback method, which can most clearly be observed for strong attacks (i.e., using high $QP_a$ values).

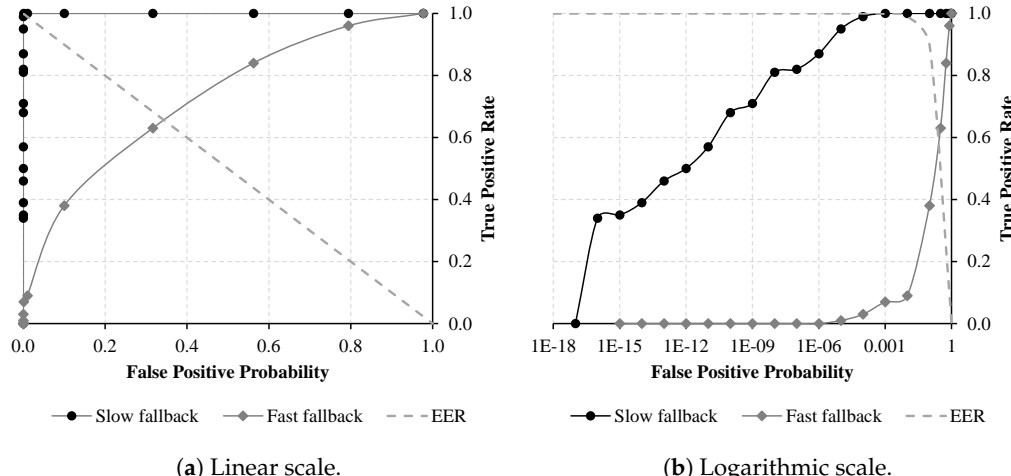

(**a**) Linear scale.  (**b**) Logarithmic scale.

**Figure 6.** ROC curves of slow and fast fallback, for $QP_w = 22$ and $QP_a = 47$, on a (**a**) linear scale and (**b**) logarithmic scale.

The robustness level of the proposed fast fallback method is not as high as the slow fallback in previous work, which is as expected since it uses much less information and is hence much faster (as thoroughly analyzed in Section 4.4). It should be noted that the robustness results can be further improved using longer video sequences. In Table 3, downsampled video segments of 10 s are used. In contrast, consider the 50-s video segment that is a concatenation of the five 10-s tested sequences (after downsampling them to a fixed framerate of 10 fps). Table 4 gives the robustness results when using $d = 4$, and $b = 16$ or $b = 8$. It can be observed that the robustness is better than in Table 3, when using 10-s segments. However, the hashes are also five times larger. Thus, again, there is a clear trade-off between robustness and storage. In general, using the fallback method demonstrates a very high level of robustness, considering that we only use secondary watermarks and the primary watermark is completely removed.

**Table 4.** Robustness results for the proposed fast fallback method using perceptual hashes, as well as the size of the perceptual hashes in bytes (B), when longer 50-s sequences are used.

| $QP_w$ | | False-Negative Rate * (%) | | | | | | | | | | | |
|---|---|---|---|---|---|---|---|---|---|---|---|---|---|
| | $QP_a =$ | 22 | 27 | 32 | 37 | 42 | 47 | 22 | 27 | 32 | 37 | 42 | 47 |
| | | | | | | | $d = 4$ | | | | | | |
| | | $b = 16$ (22,800 B) | | | | | | $b = 8$ (100,130 B) | | | | | |
| 22 | | 0 | 0 | 0 | 30 | 100 | 100 | 0 | 0 | 0 | 0 | 90 | 90 |
| 27 | | 0 | 0 | 0 | 0 | 35 | 100 | 0 | 0 | 0 | 0 | 0 | 65 |
| 32 | | 0 | 0 | 0 | 0 | 0 | 50 | 0 | 0 | 0 | 0 | 0 | 0 |
| 37 | | 0 | 0 | 0 | 0 | 0 | 0 | 0 | 0 | 0 | 0 | 0 | 0 |

* A 0% FNR means correct detection of all watermarks in all tested sequences.

### 4.4. Time Measurements

In this section, the time is measured and compared for the slow and fast fallback methods, when using hashes of various sizes. To perform all time measurements, the *Cactus* sequence is used, which is a 10-s video with resolution 1920 × 1080 and has 500 frames. Moreover, the calculation of correlation coefficients of uncompressed secondary watermark signals, the perceptual hashing, and the accuracy calculation of perceptual hashes are all implemented in Python.

As a baseline measurement, the slow fallback method of our previous work [4] requires 13,797 milliseconds (ms), i.e., approximately 13.8 s, to calculate the correlation coefficient between two 10-s watermarked videos (each approximately 1483 MB large in an

uncompressed form). The remaining time measurements in this section will be compared to this baseline measurement.

The left-hand side of Table 5 shows the time measurements to calculate the perceptual hash of a 10-s video. As described in Section 4.1, the parameters $b$ and $d$ are varied while the others are fixed. The perceptual hashing time varies between 204 ms and 1612 ms, which is approximately 9 to 68 times faster than a single correlation coefficient calculation. It can be observed that using a smaller block size $b$ and extracting more DCT coefficients (i.e., a higher $d$) results in a longer time to calculate the hash. This is as expected, since this also results in a longer hash or larger file size, as well as a higher robustness level, as observed in Table 3.

It should be stressed that the perceptual hashing can take place offline, *before* watermark detection, i.e., the hashes can be calculated when creating the watermarked videos. In this way, only the small perceptual hashes need to be stored, rather than the large watermarked videos. Then, upon watermark detection in a leaked video, only the accuracy between the perceptual hashes needs to be calculated.

**Table 5.** Time measurements of perceptual hashing and of calculating the accuracy between two perceptual hashes.

| | | Time (ms) | | | | | | |
|---|---|---|---|---|---|---|---|---|
| | | **Perceptual Hashing** | | | | **Accuracy Calculation** | | |
| $b$ | $d =$ | **0** | **2** | **4** | $d =$ | **0** | **2** | **4** |
| 16 | | 204 | 233 | 429 | | 0.15 | 0.16 | 0.25 |
| 8 | | 591 | 760 | 1612 | | 0.21 | 0.25 | 0.53 |

The largest speed-up compared to calculating the correlation coefficient between two uncompressed watermarked videos can be observed when measuring the time to calculate the perceptual hash accuracy. The right-hand side of Table 5 shows the time measurements of calculating the accuracy between two perceptual hashes, using the same varying parameters $b$ and $d$. The time measurements vary between 0.15 ms and 0.53 ms, which is approximately 26,000 to 92,000 times faster than calculating the correlation coefficient (13,797 ms). These practically observed speed-ups can be compared to the theoretical speed-up calculated in Section 3.3. When $b = 8$ and $d = 4$, the theoretically calculated speed-up is approximately 52,000, whereas the practically observed speed-up is approximately 26,000 in that case. In other words, the practical and theoretical speed-up differ by a constant factor of 2. This is not unexpected, because the asymptotic complexities ignore constant factors.

Again, it can be observed that using a smaller block size $b$ and extracting more DCT coefficients (i.e., a higher $d$) results in a longer time to calculate the accuracy. However, in any case, the accuracy calculations of hashes are much faster than the correlation calculation of secondary watermark signals. Hence, these results showcase the biggest advantage of using perceptual hashes over secondary watermark signals.

## 5. Limitations

Although the speed-ups observed in Section 4.4 are noteworthy, the proposed fallback method has some limitations. For clarity and completeness, this section briefly discusses these limitations.

First, the fallback method is still non-blind, which means that access to the original video is required for watermark detection. Additionally, the proposed method requires access to the perceptual hashes of all watermarked videos. By pre-computing these hashes, the computational complexity is of detection is reduced, but the storage requirements are slightly increased. Please note that the increase in storage is relatively small, especially in comparison with storing the watermarked videos in the slow fallback method. Perceptual hashes are very small in file size, and we can limit this to a selection of small video segments.

Secondly, the proposed fast method has a decrease in robustness compared to the slow fallback method of previous work, as observed in Section 4.3. However, it should be noted that the robustness is still higher than the primary watermarking method when targeted attacks are applied. Additionally, the robustness can be increased using more or longer video segments.

Thirdly, watermarked videos that were initially compressed using low $QP_w$ values are less robust than those compressed with high $QP_w$ values. This is because a low $QP_w$ results in fewer compression artifacts, and hence a weaker secondary watermark signal. As a result, the proposed fallback method does not work for videos that are compressed lossless or with an extremely high quality setting (i.e., a very low $QP_w$).

Fourthly, the fallback method only works for videos that are watermarked in the uncompressed domain, and that are compressed before being distributed to the users. This is also a limitation, since compressing every watermarked video is not very scalable. However, the watermarking and compression can be made faster in practice, for example using a fast compression architecture [13] or using the A/B watermarking framework [14].

## 6. Practical Example

This section briefly describes how the proposed method can be applied in practice.

To embed the watermarks, an existing primary watermarking method is applied as usual. In the experimental setup of this paper, we used the DTCWT method of Asikuzzaman et al. [10], although it should be noted that other watermarking methods that operate in the uncompressed domain can be applied as well.

Then, the watermarked videos are compressed before being distributed to the users. Ideally, the perceptual hashes are calculated and stored during this stage. To decide the number and length of video segments that are hashed, a trade-off is made between increasing the robustness on the one hand, and lowering the complexity and storage overhead on the other. As an example, if one wants to reach a robustness level as in Table 4, 50 s of video segments should be hashed. In a 1-h movie, this could translate to hashing a 10-s video segment after every 12 min of video. As another example, all video segments in the entire video could be hashed, resulting in a very high robustness level, yet in a much larger hash size.

When a video is leaked (and potentially attacked) on the Internet, it is downloaded and watermark detection can take place. First, the primary watermark detection method can be applied. If that fails, fallback detection is necessary. This can be quickly done using the fast detection using perceptual hashes proposed in this paper, i.e., the video segments are selected (either manually or automatically), and the perceptual hash is calculated. Additionally, the hashes from the watermarked video need to be loaded, which were ideally pre-computed during the compression stage. Then, the perceptual hash of the leaked video is compared to the hash of all watermarked videos, resulting in a decision of each watermark's presence. The watermark that is detected can then be linked to the culprit that leaked the video.

Only in the case that a very strong (e.g., quality-decreasing) attack is applied, and the fast fallback fails to detect a watermark, one can resort to the slow fallback method of our previous work [4].

## 7. Discussion

Existing watermarking methods have a limited robustness level. In our previous work, a fallback detection system was proposed that improves the robustness of uncompressed-domain watermarking methods that are applied before compression, with the main disadvantage of being slow. Therefore, this paper proposed to speed up the detection using perceptual hashes instead of uncompressed secondary watermark signals. We theoretically analyzed the computational complexity, and demonstrated that the detection procedure results in a practical speed-up of a factor between 26,000 to 92,000, depending on the number of bytes used to store the perceptual hash. This speed-up is observed when calculating

the hashes offline, prior to detection. In this way, an additional advantage of the proposed fast detection method is that it does not require the watermarked videos. Instead, storing the original unwatermarked video and the small perceptual hashes is sufficient. Although the robustness decreases compared to using secondary watermark signals, a high level of robustness can still be achieved, especially when the watermarked videos are initially compressed with a lower quality setting. Most importantly, it is still much more robust to targeted attacks than the primary watermarking method.

To place the time measurements in context, assume that 1000 watermarked videos are distributed to 1000 users, and one of these watermarked videos is leaked. The slow fallback watermark detection method requires approximately 4 hours to compare the leaked video to all watermarked videos. In contrast, the fast fallback detection method only requires 530 ms, i.e., approximately half a second. In addition, the fast fallback method requires the perceptual hashes to be calculated in approximately half an hour, but this can be done offline, prior to detection, e.g. when creating and distributing the watermarked videos.

Future work should investigate if the proposed fast secondary watermark detection method can also be applied to other primary watermarking techniques that are based on similar underlying concepts. For example, it can be applied to other methods that represent a (primary) watermark as a collection of compression artifacts [19,22]. Additionally, future research could be dedicated to examining the performance of other perceptual hashing methods to perform fallback detection, and to making the fallback method blind.

**Author Contributions:** All authors took part in the discussion of the work described in this paper. All authors have read and agreed to the published version of the manuscript.

**Funding:** This work was funded in part by the Research Foundation—Flanders (FWO) under Grant 1S55218N, in part by IDLab (Ghent University—imec), in part by Flanders Innovation & Entrepreneurship (VLAIO) project RHETORiC (HBC.2019.0055), and in part by the European Union.

**Data Availability Statement:** The results presented in this study are available on http://media.idlab.ugent.be/?p=1264 accessed on 12 May 2021.

**Acknowledgments:** The computational resources (STEVIN Supercomputer Infrastructure) and services used in this work were kindly provided by Ghent University, the Flemish Supercomputer Center (VSC), the Hercules Foundation and the Flemish Government department EWI.

**Conflicts of Interest:** The authors declare no conflict of interest. The funders had no role in the design of the study; in the collection, analyses, or interpretation of data; in the writing of the manuscript, or in the decision to publish the results.

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
