# Peer review of "Fast Fallback Watermark Detection Using Perceptual Hashes"

_electronics, doi:10.3390/electronics10101155_

Round 1

Reviewer 1 Report

The authors described Fast Fallback Watermark algorithm using perceptual hashes. However, in the description of the submitted article, I noticed a few shortcomings that should be corrected in this article to get complete pictures of the described method.

1. Trade-offs in watermarking systems is related to the following parameters: the specific perceptual transparency achieved, the robustness, and the information capacity (payload) obtained. Please consider the information capacity description in the article and what information capacity the developed method will provide? It is best to calculate the capacity (payload) using formulas or analytical calculations. On what parameters of the described algorithm depends the output information capacity?

2. To assess the quality of the watermarking system, the authors should consider determining the standard ROC operating curves in such cases. The ROC curves show the distribution of the first and second type errors and show the intersection of the curves for these errors named Equal Error Rate (EER).

3. Moreover, there was no presentation of the results of the perceptual transparency watermarked video in the form of objective fidelity results, e.g. PSNR, NC, etc., video degradation compared to the original host without watermarking from the receiving side.

Author Response

We appreciate the time you took to read, analyze and comment the work we provided you. By addressing your comments in our revised manuscript, we believe that the paper is more clear and the quality of our work improved. In the following, we reply to each of your comments individually.

1. Trade-offs in watermarking systems is related to the following parameters: the specific perceptual transparency achieved, the robustness, and the information capacity (payload) obtained. Please consider the information capacity description in the article and what information capacity the developed method will provide? It is best to calculate the capacity (payload) using formulas or analytical calculations. On what parameters of the described algorithm depends the output information capacity?

It is indeed true that typical watermarking systems are evaluated using a trade-off between the perceptibility, robustness and payload capacity. However, we would like to note that our paper proposes a fallback detection method, rather than a complete watermark embedding-and-detection system. Hence, the proposed fallback method has no direct control over watermark embedding. Instead, we apply an existing primary watermark embedding method from the state of the art, and show that we can alternatively detect it using a secondary watermark.

For the evaluation of our paper, we used the DTCWT method to perform primary watermark embedding. This method embeds a zero-bit watermark in the video, which means that a watermark seed is used to pseudorandomly generate a watermark signal that is embedded in certain DTCWT coefficients. The pseudorandom signal does not hold any specific payload, i.e., it is not a multi-bit message. Instead, it is a zero-bit watermark, which means that its presence can be detected and linked to the seed that was used to generate it.

In a similar way, the secondary watermark of our proposed method is considered a zero-bit watermark. This means that only its presence or absence can be detected and linked to a seed (which can then be linked to a user, timestamp, etc.). The zero-bit watermark can hence not be used to embed a specific multi-bit message. This also means that the parameters of our proposed method do not influence the information capacity.

We have clarified that the watermark is a zero-bit watermark in Section 3.2. Fast Fallback Detection using Perceptual Hashes of the revised manuscript.

2. To assess the quality of the watermarking system, the authors should consider determining the standard ROC operating curves in such cases. The ROC curves show the distribution of the first and second type errors and show the intersection of the curves for these errors named Equal Error Rate (EER).

Thank you for your suggestion. In the revised paper, we have expanded the robustness evaluation in Section 4.3. Robustness. More specifically, the ROC curves shown in Figure 6 compare the performance of the slow and fast fallback method. Additionally, the figure also plots the EER line to show the intersections of the errors.

3. Moreover, there was no presentation of the results of the perceptual transparency watermarked video in the form of objective fidelity results, e.g. PSNR, NC, etc., video degradation compared to the original host without watermarking from the receiving side.

We would again like to note that our proposed fallback method has no direct control over watermark embedding. The perceptibility of the primary watermark is only influenced by the (existing) primary watermarking method. Additionally, the perceptibility of the secondary watermark is only influenced by the video encoder that compresses the watermarked video.

For completeness, we have added a section that discusses the perceptibility of the primary and secondary watermark (Section 4.2 Perceptibility and Table 1). First, we show that the primary watermark results in average PSNR of 37.92 dB in the U channel, and a PSNR of infinity in the Y channel (since the primary watermark is embedded only in the U channel). Second, we measure the perceptibility of the secondary watermark in the Y channel, which are the compression artifacts created during video encoding. Additionally, we demonstrate that the secondary watermark perceptibility is approximately equal to the perceptibility of the compression artifacts that are created during compression of an unwatermarked video. Hence, the secondary watermark used by the proposed fallback method does not degrade the video quality more than ordinary unwatermarked compression.

Reviewer 2 Report

The authors present a fast technique for detecting watermarks in a video track. My comments on the work:
- a computational complexity analysis of the method should be added to the paper,
- it is worth to add a section of critical analysis and in it to describe limitations and difficulties in the application of methods, what kind of transformations make difficult the application of the developed technique,
- if possible, add to the paper a toy example presenting the whole process of applying the technique in practice,
Apart from the above minor corrections, I consider the work complete and worth publishing.

Author Response

We are happy you consider the work complete and worth publishing, apart from some minor corrections. We appreciate the time you took to read, analyze and comment the work we provided you. We have addressed your concerns in the revised manuscript and believe it improved the quality of our work. In the following, we reply to each of your comments individually.

1. A computational complexity analysis of the method should be added to the paper.

Thank you for pointing out that a formal complexity analysis is missing. In the revised paper, we added Section 3.3. Theoretical Complexity Analysis, which discusses the asymptotical computational complexity of slow and fast watermark detection. Additionally, in Section 4.4. Time Measurements of the revised paper, we added a comparison between the practically observed speed-up with the theoretically expected speed-up.

2. It is worth to add a section of critical analysis and in it to describe limitations and difficulties in the application of methods, what kind of transformations make difficult the application of the developed technique.

Thank you for your valuable suggestion. In the revised paper, we added Section 5. Limitations, which lists and discusses the main limitations of the fast fallback method.

3. If possible, add to the paper a toy example presenting the whole process of applying the technique in practice.

Thank you again for your valuable suggestion. In the revised paper, we added Section 6. Practical Example. This section describes how the whole process of embedding, compression and fallback detection can be applied in practice.

Round 2

Reviewer 1 Report

I have checked all corrections.  All required corrections were carried out in this research.  The paper presents a fine level of research description and can be published in the MDPI SENSORS journal.

Author Response

Thank you for taking your time to check the revised paper. We are happy to hear that you are satisfied with the corrections we carried out, and that you recommend to accept our paper for publication.